# Factors associated with interobserver variation amongst pathologists in the diagnosis of endometrial hyperplasia: A systematic review

Chloe A. McCoy[1]*, Helen G. Coleman[1], Charlene M. McShane[1], W. Glenn McCluggage[2], James Wylie[3], Declan Quinn[3], Úna C. McMenamin[1]

1 Centre for Public Health, Queen's University Belfast, Belfast, Northern Ireland, United Kingdom,
2 Department of Pathology, Belfast Health and Social Care Trust, Belfast, Northern Ireland, United Kingdom,
3 Department of Obstetrics and Gynaecology, Antrim Area Hospital, Northern Health and Social Care Trust, Antrim, Northern Ireland, United Kingdom

* cmccoy16@qub.ac.uk

## Abstract

### Objective

Reproducible diagnoses of endometrial hyperplasia (EH) remains challenging and has potential implications for patient management. This systematic review aimed to identify pathologist-specific factors associated with interobserver variation in the diagnosis and reporting of EH.

### Methods

Three electronic databases, namely MEDLINE, Embase and Web of Science, were searched from 1st January 2000 to 25th March 2023, using relevant key words and subject headings. Eligible studies reported on pathologist-specific factors or working practices influencing interobserver variation in the diagnosis of EH, using either the World Health Organisation (WHO) 2014 or 2020 classification or the endometrioid intraepithelial neoplasia (EIN) classification system. Quality assessment was undertaken using the QUADAS-2 tool, and findings were narratively synthesised.

### Results

Eight studies were identified. Interobserver variation was shown to be significant even amongst specialist gynaecological pathologists in most studies. Few studies investigated pathologist-specific characteristics, but pathologists were shown to have different diagnostic styles, with some more likely to under-diagnose and others likely to over-diagnose EH. Some novel working practices were identified, such as grading the "degree" of nuclear atypia and the incorporation of objective methods of diagnosis such as semi-automated quantitative image analysis/deep learning models.

**Data Availability Statement:** All relevant data are within the paper and its Supporting Information files.

**Funding:** CMcC is funded by a Department for the Economy Studentship https://www.nidirect.gov.uk/articles/department-economy-postgraduate-studentship-scheme. ÚMcM is funded by a UKRI Future Leaders Fellowship https://www.ukri.org/what-we-do/developing-people-and-skills/future-leaders-fellowships/. HC is supported by a Cancer Research UK Career Establishment Award (C37703/A25820) https://www.cancerresearchuk.org/funding-for-researchers/our-funding-schemes/career-establishment-award. The funders had no role in study design, data collection and analysis, decision to publish, or preparation of the manuscript.

**Competing interests:** The authors have declared that no competing interests exist.

## Conclusions

This review highlighted the impact of pathologist-specific factors and working practices in the accurate diagnosis of EH, although few studies have been conducted. Further research is warranted in the development of more objective criteria that could improve reproducibility in EH diagnostic reporting, as well as determining the applicability of novel methods such as grading the degree of nuclear atypia in clinical settings.

## Introduction

Endometrial cancer (EC) is the most common gynaecological malignancy in developed countries, accounting for over 417,000 cases and 97,000 deaths worldwide in 2020 [1]. EC incidence rates have increased rapidly over the last few decades, particularly in high-income countries, likely due to rising obesity rates, greater life expectancy, and changes in reproductive patterns [2,3]. Endometrial hyperplasia (EH) is a recognised precursor lesion of EC [4], specifically the most common endometrioid type, and its accurate and early detection offers opportunities for optimal patient management and cancer prevention.

Over a 20-year period, the cumulative risk of EC development has been estimated to be less than 5% for patients diagnosed with EH without atypia, rising to 28% for patients diagnosed with atypical hyperplasia [5]. The accurate distinction between EH without atypia and atypical hyperplasia, and between atypical hyperplasia andEC, remains a challenging area in diagnostic pathology and has important consequences for patient management [6]. Hysterectomy with bilateral salpingo-oophorectomy is the recommended treatment options for atypical EH [7]. However, for premenopausal women who wish to preserve fertility or patients who are a poor operative risk, progestogen therapies with close surveillance may be recommended to avoid or delay surgery [7]. Consequently, an erroneous diagnosis may lead to patients undergoing multiple endometrial biopsies, and/or under- or over-treatment [6,8].

Interobserver variation in the diagnosis of EH has been well described in the literature [6,9–13]. Some histopathological challenges attributable to variation in diagnosis include, but are not limited to, inadequate tissue and specimen fragmentation, the use of hormonal therapies that can "mask" features, and the fact that EH can be a focal lesion [6,14]. The diagnostic criteria for EH have been continuously adapted and refined in an effort to overcome interobserver variation. Previously, the most widely used criteria was the World Health Organisation (WHO) 1994 classification system which categorised EH into four groups according to the complexity of glandular architecture and the presence or absence of cytologic atypia [4]. However, diagnostic reproducibility using this system was poor, in part due to significant variability in the assessment of nuclear atypia [6]. The endometrioid intraepithelial neoplasia (EIN) system was introduced in 2000 and was developed based on correlation of morphological features with clinical outcome, molecular changes, and objective computerised histomorphometry [8,15]. A subsequent update to the WHO criteria in 2014 simplified the classification into two groups; (i) EH without atypia and (ii) atypical hyperplasia based upon the absence or presence of cytological atypia, with the diagnosis of EIN in this system considered largely interchangeable with atypical hyperplasia [16,17]. In 2020, the WHO criteria were unchanged, although the value of biomarkers to enhance diagnosis was discussed, including the loss of immunoreactivity with PTEN, PAX2 and mismatch repair proteins and the nuclear expression of beta-catenin [6,18]. However, despite evolving classification criteria, interobserver variation between pathologists in EH diagnosis is still significant [6], and thus, a thorough

understanding of factors influencing this variability is required to help improve quality assurance in EH diagnostic pathology [19].

It is currently unclear whether pathologist-specific characteristics, such as speciality, training or working environment can be attributed to interobserver variation in the diagnosis of EH. Furthermore, it is relatively unknown whether specific working practices or the methods used for EH diagnosis impacts on overall diagnostic reproducibility, including more novel practices such as the use of artificial intelligence or biomarkers. Therefore, the aim of this systematic review was to identify pathologist-specific factors associated with interobserver variation in the diagnosis and reporting of EH.

## Methods

This systematic review was reported in line with Preferred Reporting Items for Systematic Reviews and Meta-Analyses (PRISMA) guidelines [20] (see **S1 Checklist**), and the protocol was registered with PROSPERO (PROSPERO 2022: CRD42022309957) [21]. Three electronic databases, MEDLINE (US National Library of Medicine, Bethesda, Maryland, USA), Embase (Reed Elsevier PLC, Amsterdam, Netherlands) and Web of Science (Thompson Reuters, Times Square, New York, USA) were systematically searched using key words and relevant medical subject headings to identify relevant studies published between 1st January 2000 and 25th March 2023 (see **S1** and **S2 Appendices**). Eligible studies had to apply the WHO 2014/ WHO 2020 classification, or the EIN 2000 classification for EH diagnoses. Therefore, the search was limited to studies from the year 2000 onwards, and no language restriction was applied.

### Inclusion criteria

Covidence was used to manage the removal of duplicates and screening process. Title and abstract screening was conducted independently by at least two reviewers (CMcC, ÚMcM, HC or CMcS) against the eligibility criteria. Full texts were then independently screened by at least two reviewers to identify studies that aligned with the inclusion criteria below:

i. **Population:** Pathologists (excluding trainees) who have reported an EH diagnosis

ii. **Intervention(s):** The diagnostic assessment of EH specimens

iii. **Comparators:** Pathologist characteristics, experience, training, working environment and working practices, e.g., the classification system used, the use of immunohistochemical biomarkers, the use of digital versus glass review, double-reporting and any other factors reflective of pathologist characteristics not listed above

iv. **Outcome:** Interobserver variation in pathologist diagnosis of EH

All studies assessing interobserver variation in the diagnostic assessment of EH in endometrial specimens (biopsy and hysterectomy samples) by pathologists were included. Studies were included if they reported interobserver agreement using Cohen's or Fleiss' Kappa statistic (κ) and/or percentage agreement, or if sufficient information was provided to calculate these. Additional outcomes considered included interobserver variation in the differentiation between EH and EC diagnoses, and any intraobserver variation outcomes if reported. Review articles, animal studies, articles that used the 1994 WHO criteria for EH classification, and studies that included only EC specimens were excluded. The reference lists of included studies were manually searched for additional articles. Any discrepancies throughout the review process were resolved through discussions with a third author.

## Data extraction

Data extraction was performed by two authors (CMcC and HC), and the following data was extracted from included studies where possible: author name, year of publication, study type and institution; the sampling method; the number of reviewing pathologists included in the study and related characteristics where present (country of practice, practice setting, time in practice and level of experience, number of biopsies reported monthly/annually); pathologist working practices (i.e., method used for analysing specimens) and the classification criteria used for EH diagnosis.

## Data synthesis

A meta-analysis was not conducted due to the heterogeneity of the study designs, and so a narrative synthesis was performed according to Synthesis Without Meta-analysis (SWiM) guidelines [22]. Included studies were ordered by year of publication and the certainty of evidence was evaluated based on the number of reviewing pathologists and the risk of bias. In some cases, data had to be transformed, i.e., the manual calculation of percentage agreement based on the data included within the study.

## Risk of bias assessment

The risk of bias within individual studies was assessed using the Quality Assessment of Diagnostic Accuracy Studies 2 (QUADAS-2) tool [23]. This assessed the risk of bias through four main domains: (1) Patient selection (low risk if consecutive patients were included and/or appropriate exclusions were considered, e.g., patients with a pre-hysterectomy diagnosis of EC); (2) Index test (low risk if reviewing pathologists were blinded to clinical information and/or the results of the reference standard, e.g., review by an expert panel who agreed consensus); (3) Reference standard (low risk if the cases included were likely to have been correctly classified and if the distributions of the cases included would likely be encountered in clinical practice); (4) Flow and timing (low risk if all reviewing pathologists received the same reference standard). In each domain, the studies were categorised as low, unclear, or high risk of bias. The QUADAS-2 tool also considers concerns of applicability, which enables the first three domains to be further assessed based on whether the criteria of the individual study was appropriate, but overall did not fit the main objectives of our systematic review [23]. One author (CMcC) performed the quality assessment and risk of bias analysis.

# Results

The study selection process is summarised in the PRISMA flow chart in **Fig 1**. Following the removal of duplicates, the titles and abstracts of 2,515 articles were independently screened by at least two reviewers. A total of 63 articles were identified for full-text review, of which eight articles met the inclusion criteria [24–31].

The characteristics of the included studies are highlighted in **Table 1.** Publication year ranged from 2005 to 2022, and the majority of studies were conducted in Europe and the USA. Five of the included studies were multi-centre [26,28–31], with the remaining three studies all single centre [24,25,27]. The majority of included studies had less than 10 reviewing pathologists, although two larger studies included 20 and 78 reviewing pathologists respectively [29,30]. Extensive pathologist characteristics in conjunction with overall agreement could only be extracted from two studies [29,30], with the other studies providing minimal detail regarding pathologist speciality, location or practice settings. Some novel working practices/methods were assessed in some of the studies, such as grading the degree of nuclear atypia [26], and subjective diagnosis in addition to/in comparison with objective methods of diagnosis (semi-automated quantitative image analysis, deep learning models) [24,25], as shown in **Table 1**. One

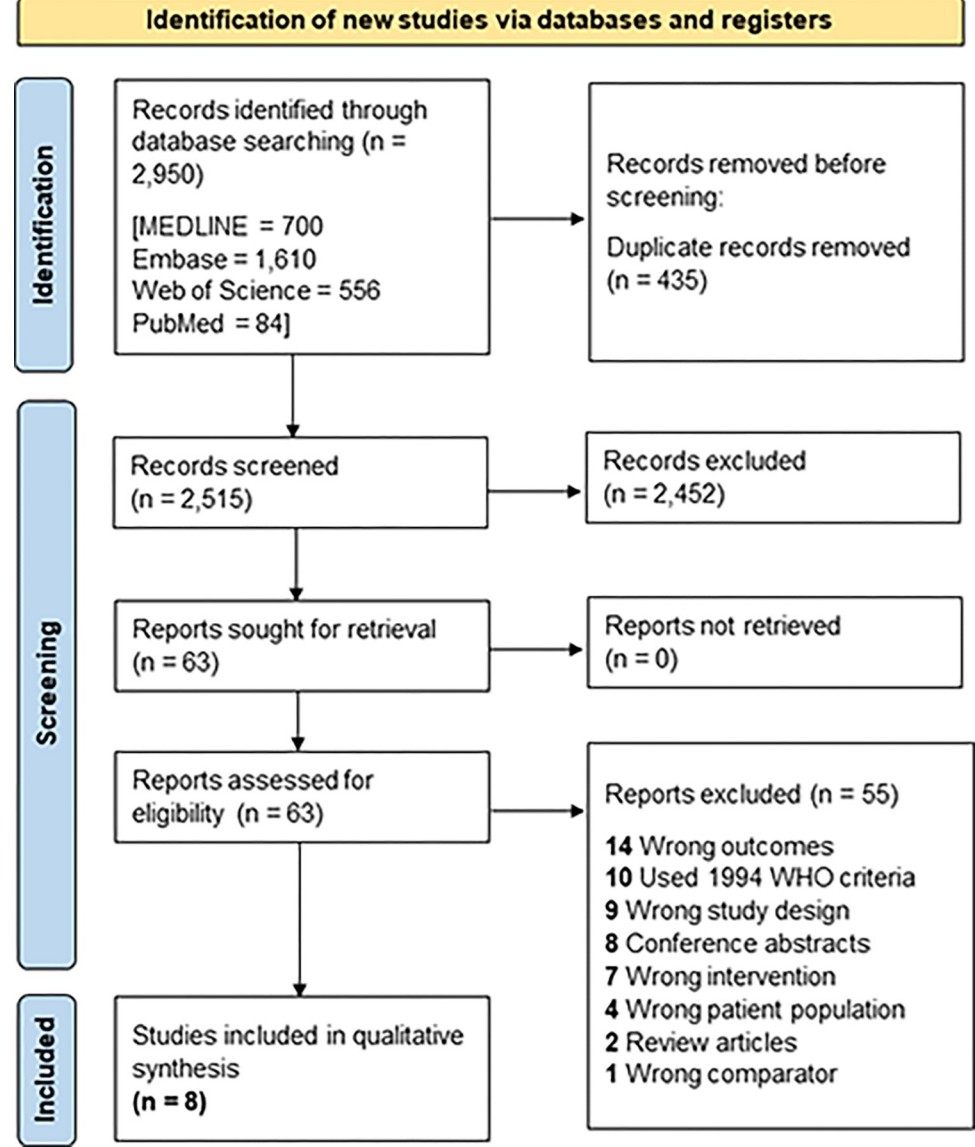

**Fig 1. PRISMA flow diagram demonstrating the study selection process.**

study assessed pathologist diagnosis using both the EIN and WHO criteria but simplified the WHO categorisation into two categories, benign hyperplasia or atypical hyperplasia/grade 1 EC [28]. However, classifying both atypical hyperplasia and EC into the one diagnostic category was not useful for the scope of our review; therefore, only the results from the pathologist diagnosis using the EIN system was extracted.

## Quality assessment

The results of the risk of bias assessment are shown in **Fig 2**. All studies were classed as a "low risk" of bias in the patient selection domain. For the index test, two studies were highlighted as having an "unclear" risk as it was uncertain whether the reviewing pathologists were blinded to the results of the reference standard [27,31]. In the reference standard and flow and timing domains, all included studies were classed as having a "low risk" of bias. Applicability concerns

**Table 1. Characteristics of the included studies.**

| Author and year | Country | Study type | Institution | Specimens | Number of reviewing pathologists | Pathologist characteristics | Pathologist working practices | Classification criteria used |
|---|---|---|---|---|---|---|---|---|
| Zhao et al., 2022 [24] | China | Single centre | Department of Pathology of Northwest Women's and Children's Hospital | Curettage, hysteroscopic surgery or hysterectomy specimens (n = 602) | 3 | Years of experience, speciality | Subjective histopathological diagnosis in comparison to using a global-to-local multi-scale convolutional neural network | WHO 2014 criteria |
| Sanderson et al., 2022 [25] | UK | Single centre | Pathology Department of the NHS Lothian Health Board | Archived endometrial hyperplasia samples (n = 125) | 2 | Speciality | Subjective histopathological diagnosis; objective diagnosis using digital computerised quantitative image analysis; use of immunohistochemical staining to aid diagnosis | WHO 2014 criteria |
| D'Angelo et al., 2021 [26] | Spain/ Italy | Multi-centre | Hospital de la Santa Creu i Sant Pau, Barcelona, Spain; Hospital Santa Chiara, Trento, Italy | Biopsies and hysterectomies (n = 79) | 3 | Location and speciality | Biopsy versus hysterectomy diagnosis of endometrial hyperplasia specimens from the same patient, according to the degree and grade of nuclear atypia | WHO 2014 criteria |
| Spoor and Cross., 2019 [27] | UK | Single centre | Department of Cellular Pathology, Queen Elizabeth Hospital | Biopsies and hysterectomies (n = 630) | 7 | Speciality | Central pathology review to determine the concordance between a) original referral histology with review histology and b) final review histology with hysterectomy histology | WHO 2014 criteria |
| Ordi et al., 2014 [28] | Spain | Multi-centre | University of Granada | Biopsies and curettages (n = 196) | 9 | Location and speciality | Subjective histopathological diagnosis | EIN criteria |
| Usubutun et al., 2012 [29] | Turkey | Multi-centre | Pathology Department of the Hacettepe University | Biopsies and curettages (n = 62) | 20 | Location; years of experience; pathology speciality; training institution; practice institution; diagnostic style group | Number of endometrial biopsies signed-out per month; current criteria used in daily practice | EIN criteria |
| Marotti et al., 2011 [30] | USA | Multi-centre | Beth Israel Deaconess Medical Centre | Biopsies (n = 18) | 78 | Location; current position; time in practice; practice setting | Number of endometrial curettings per month; current criteria used in daily practice | EIN criteria |
| Hecht et al., 2005 [31] | USA | Multi-centre | Beth Israel Hospital Department of Pathology | Biopsies and curettages (n = 97) | 3 | Practice setting and speciality | Subjective histopathological diagnosis; morphometric diagnosis using computerised morphometry (D-score) | EIN criteria |

were raised in one study for the index test domain, as the study provided little information on how this was conducted [27].

## Pathologist-specific characteristics and interobserver agreement

A total of 125 reviewing pathologists were included across the eight studies, and most studies included at least one specialist gynaecological pathologist (see **Table 2**). Marotti et al. assessed

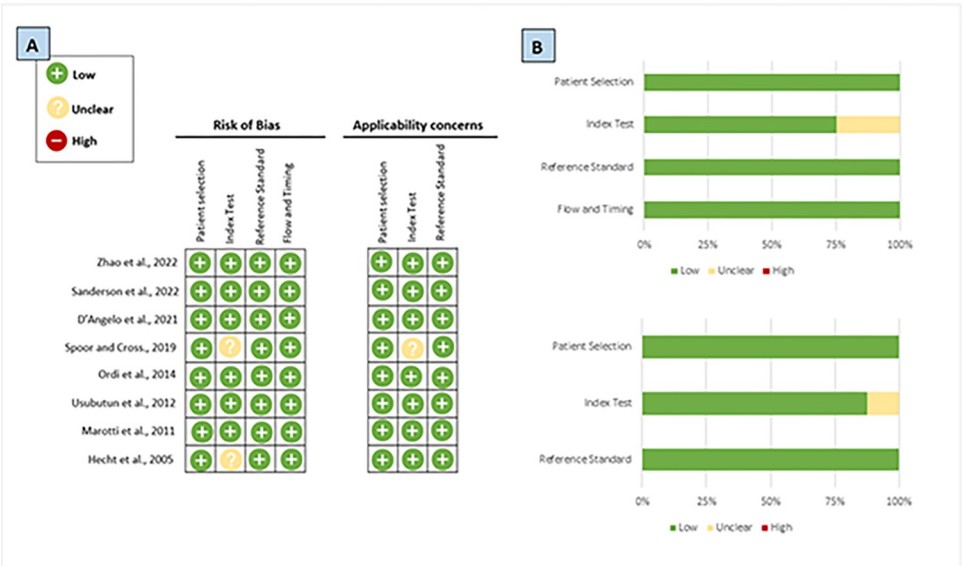

**Fig 2. The risk of bias assessment using the QUADAS-2 tool.** (A) Summary of the risk of bias and applicability concerns across the individual studies. (B) Graphical summary of the risk of bias and applicability concerns displayed as percentages.

the reproducibility of the EIN criteria amongst 78 pathologists across 13 different countries [30]. They reported that pathologists with a specialist interest in gynaecological pathology had better levels of agreement with the reference standard (62%), compared to pathologists with other main interests (57%) and pathology residents (47%). Two specialist gynaecological pathologists had 100% agreement in this study; both had been practicing for over ten years, diagnosed more than 20 EH specimens per month, and used the EIN classification system in their daily practice [30]. However, in the study of 20 reviewing European pathologists by Usubutun et al., years in practice did not appear to influence overall agreement, as similar levels of agreement were found in those with three to ten years compared to ten plus years of experience ($p = 0.529$) [29]. Both studies provided pre-reading and/or training modules to help the reviewing pathologists unfamiliar with the EIN criteria, and Marotti et al. reported those utilising this information had higher levels of concordance with the reference standard (60%) than those who did not (40%). In terms of the terminology used in daily practice, in both studies the majority of pathologists typically used the WHO terminology; however, this did not appear to impact concordance with the reference standard when using the EIN criteria. Furthermore, Usubutun et al, reported no statistical association between practice type ($p = 0.926$), training institution ($p = 0.082$), practice institution ($p = 0.255$), or classification system used in daily practice ($p = 0.437$) with the extent of overall concordance [29]. Usubutun et al. further categorised 20 reviewing pathologists into groups (red, yellow and green) based on their personal diagnostic style. Most of the pathologists (yellow style group, n = 11) tended to diagnose cases in a balanced spectrum, whereas others were more likely to diagnose benign lesions more so than EIN or EC (green style group, n = 4), and some were more likely to diagnose EIN (red style group, n = 5) than benign lesions. The red style group had the highest level of concordance (83.2%; κ = 0.71–0.83), the yellow group slightly lower (81.4%, κ = 0.66–0.82), and the green group displayed the lowest levels of concordance (70.8%; κ = 0.45–0.68) [29]. In terms of specific characteristics, personal diagnostic style was not statistically associated with years of experience ($p = 0.435$), practice type ($p = 0.228$), training institution ($p = 0.236$), practice institution ($p = 0.204$), or classification system used ($p = 0.376$) [29].

**Table 2. Factors associated with interobserver variation amongst pathologists in the diagnosis of endometrial hyperplasia across the eight included studies.**

| Author and year | Interobserver agreement between pathologists | | | Summary of findings | Comments |
|---|---|---|---|---|---|
| | Comparison | Cohens/ Fleiss κ | % Agreement | | |
| Zhao et al., 2022 [24] | Normal endometrium | n.r. | 99.4–100% | *Years of experience*: <br>• Junior pathologist, two years: n = 1 <br>• Mid-level pathologist, six years: n = 1 <br>• Senior pathologist, 15 years: n = 1*Main points*:Overall accuracy between diagnostic categories <br>• For EIN, Pathologist 1 (junior) had the lowest diagnostic accuracy (74.9%), with Pathologist 3 (senior) having the highest diagnostic accuracy (98.1%) <br>• G2LNet had the highest accuracy when diagnosing cases of EIN (99.8%), performing better than all three pathologists <br>• G2LNet was better at diagnosing cases of hyperplasia without atypia (85.8%) compared to Pathologist 1 (82.8%), although was less accurate than Pathologists 2 (93.5%) and 3 (97.8%) <br>*Comparison between pathologist diagnosis and G2LNet*: <br>• Accuracy ranged from 85–98.6% between the pathologists, GL2Net had 95.3% accuracy <br>• Kappa values for agreement between pathologist diagnosis and G2LNet ranged from 0.72–0.93, with the most senior pathologist having the higher Kappa | The deep learning model was superior when diagnosing the premalignant lesion (atypical EH/EIN), whereas pathologists were better at diagnosing normal endometrium and hyperplasia without atypia. This indicates that G2LNet could be used to complement pathologists in the automatic diagnosis of the precancerous endometrial lesion |
| | Hyperplasia without atypia | n.r. | 82.8–97.8% | | |
| | Atypical hyperplasia/EIN | n.r. | 74.9–98.1% | | |
| Sanderson et al., 2022 [25] | Benign endometrium | n.r | 3.2% | *Speciality*: <br>• Specialist gynaecological pathologist: n = 2 (100%) <br>*Main points*: <br>• Pathologist B was more likely to diagnose EIN (n = 66 cases) than Pathologist A (n = 46 cases) <br>• Comparison between the index cases of complex AH and those reclassified with the EIN/WHO 2014 system showed that 3 were reclassified as non-atypical EH and one as EC <br>• EIN/WHO 2014 system was accurate at predicting the absence of subsequent EC (NPV = 98.4%) <br>• Using semi-automated quantitative image analysis, 3/10 cases with a final consensus diagnosis of EIN met the EIN diagnostic criteria for a VPS of <55% whereas 7/10 cases did not show image analysis evidence of EIN <br>• All final consensus cases of non-atypical EH (n = 11) met the architectural requirements of the EIN/WHO 2014 system and had a VPS >55% <br>• Expression of p53 and MMR proteins could not distinguish between non-atypical EH and EIN; ARID1A loss (p = 0.011) and altered HAND2 expression (p <0.001) were significantly associated with an EIN diagnosis <br>• A panel of HAND2, PTEN and PAX2 was useful in identifying patterns associated with EIN and non-atypical EH based on diagnostic criteria and could be applicable in identifying those likely to have benign EH. | Application of the revised EIN/WHO 2014 is more likely to result in a consensus EIN diagnosis, although agreement between the two pathologists was only "fair" after combining diagnostic categories. Computer-aided imaging of the gland-to-stroma ratio of EH specimens can be utilised to assist pathologist diagnosis and improve diagnostic accuracy–for example, it could be useful to validate the exclusion of EIN. |
| | Non-atypical hyperplasia | n.r | 29.6% | | |
| | Atypical hyperplasia/EIN | n.r | 32.0% | | |
| | Atypical/hyperplasia/EIN in an endometrial polyp | n.r | 0% | | |
| | Hyperplastic polyp | n.r | 2.4% | | |
| | Endometrial cancer | n.r | 0% | | |
| | Combined total | 0.48 | 67.2% | | |

*(Continued)*

**Table 2.** (*Continued*)

| Author and year | Interobserver agreement between pathologists | | | Summary of findings | Comments |
|---|---|---|---|---|---|
| | Comparison | Cohens/ Fleiss κ | % Agreement | | |
| D'Angelo et al., 2021 [26] | Biopsy diagnosis (low/high-grade atypical hyperplasia) | 0.72–0.81 | 87.3–91.1% | *Location*: <br>• Spain: n = 2 <br>• Italy: n = 1 <br>*Speciality*: <br>• Gynaecological pathology: n = 3 (100%) <br>*Main points*: <br>• Recommends the introduction of a novel method of grading the degree of nuclear atypia using robust histological criteria <br>Degree of nuclear atypia was predictive of the findings at hysterectomy ($p = 1.6 \times 10^{-15}$) <br>• No patients with low-grade AH (n = 53) had EC at hysterectomy, whereas 16 patients (61%) with high-grade AH at biopsy had grade 1 EC at hysterectomy <br>• Molecular analysis of specimens suggested that AH is molecularly heterogenous | Diagnosis of AH using a binary classification of nuclear atypia was highly reproducible amongst gynaecological pathologists from three different institutions. Overall, the review of biopsies resulted in increased diagnostic concordance compared to hysterectomies. Correlation of biopsy findings with clinical data and diagnostic imaging could help improve concordance. |
| | Hysterectomy diagnosis (benign endometrium, non-atypical hyperplasia, low-grade atypical hyperplasia, high-grade atypical hyperplasia, grade 1 endometrial cancer) | 0.60–0.71 | 72.2–79.8% | | |
| Spoor and Cross., 2019 [27] | Atypical hyperplasia (original pathology biopsy report versus central review) | n.r. | 68% | *Pathologist factors*: <br>• Experienced cellular pathologists: n = 7 (100%) <br>• Report gynaecologic cancer cases on a regular basis <br>• All reviewing pathologists take part in a national gynaecologic histology external quality assurance <br>*Main points*: <br>• One of the main reasons for discrepant results was the accurate distinction between atypical EH and EC <br>• 8 cases (6.1%) in which EC was diagnosed on biopsy, but on hysterectomy there was no cancer present; 47 cases (11.6%) originally diagnosed as AH and/or Grade 1 EC that were upgraded to high-grade EC upon review | Overall agreement was increased when comparing biopsy and final hysterectomy specimens in comparison to reviewing the original pathology report. The central pathology review highlighted areas of diagnostic disagreement that could ultimately impact patient management, thus highlighting the importance of a central review process in helping improve diagnostic accuracy. |
| | Atypical hyperplasia (review biopsy opinion versus final histology at hysterectomy) | n.r. | 90% | | |
| Ordi et al., 2014 [28] | Benign cycling endometrium | 0.67 | 11.2% | *Location*: <br>• Europe: n = 5 (56%) <br>• United States: n = 4 (44%) <br>*Speciality*: <br>• Specialist gynaecological pathologist: n = 9 (100%) <br>*Other points*: <br>• 39% full agreement (κ = 0.434) was obtained using EIN criteria with four categories <br>• 58% full agreement (κ = 0.528) was obtained using EIN criteria with three categories (a. benign endometrium; b. benign hyperplasia; c. EIN and EC) <br>• 69% agreement (κ = 0.589) was obtained using EIN criteria with two categories (a. benign endometrium and benign hyperplasia; b. EIN and EC). | Complete agreement only occurred in a third of the biopsies using the EIN criteria, and interobserver variability was high even amongst expert pathologists. Reducing the number of diagnostic categories resulted in higher levels of agreement. Better reproducibility was associated with diagnosing benign endometrium or endometrial cancer. |
| | Benign hyperplasia | 0.35 | 9.7% | | |
| | EIN | 0.27 | 7.1% | | |
| | Endometrial cancer | 0.52 | 11.2% | | |
| | All groups combined | 0.43 | 39.2% | | |

(*Continued*)

**Table 2.** (Continued)

| Author and year | Interobserver agreement between pathologists | | | Summary of findings | | | Comments |
|---|---|---|---|---|---|---|---|
| | Comparison | Cohens/ Fleiss κ | % Agreement | | | | |
| Usubutun et al., 2012 [29] | Benign hyperplasia | 0.64 | n.r. | **General pathologist characteristics** *Years of pathology practice*: • <3: n = 1 (5%) • 3–10: n = 5 (25%) • >10: n = 14 (70%) *Practice type* • Gynaecological pathologist: n = 17 (85%) • General pathologist: n = 3 (15%) *Work setting*: • University: n = 16 (80%) • Public hospital: n = 4 (20%) *Number of endometrial biopsies seen per month*: • <10: n = 2 (10%) • 10–20: n = 1 (5%) • >20: n = 17 (85%) *Terminology used in daily practice*: • EIN criteria: n = 4 (20%) • WHO criteria: n = 16 (80%) *Pathologists who undertook the recommended pre-reading*: • Yes: n = 18 (90%) • No: n = 2 (10%) *Current practice institution is the same as training institution*: • Yes: n = 13 (65%) • No: n = 7 (35%) *Diagnostic style group*: • Red: n = 5 (25%) • Yellow: n = 11 (55%) • Green: n = 4 (20%) | **Pathologist characteristics according to diagnostic agreement** *Years of pathology practice*: • <3: (79% agreement; κ = 0.71) • 3–10: (78.8% agreement; κ = 0.66–0.77) • >10: (80.4% agreement; κ = 0.45–0.83) *Practice type*: • Gynaecological pathologist: (79.7% agreement; κ = 0.45–0.83) • General pathologist: (80.1% agreement; κ = 0.68–0.77) *Uses the EIN criteria in daily practice*: • Yes: (82.1% agreement; κ = 0.68–0.77) • No: (79.2% agreement; κ = 0.45–0.83) *Diagnostic style group*: • Red: (83.2% agreement; κ = 0.71–0.83) • Yellow: (81.4% agreement, κ = 0.66–0.82) • Green: (70.8% agreement; κ = 0.45–0.68) | **Statistical association** No statistical association between years of experience (P = 0.529), practice type (P = 0.926), training institution (P = 0.082), practice institution (P = 0.255), or classification system used in practice (P = 0.437) with the extent of concordance No statistical association between personal diagnostic style and years of experience (P = 0.435), practice type (P = 0.228), training institution (P = 0.236), practice institution (P = 0.204), or classification system used in practice (P = 0.376) | The reviewing pathologists had personal diagnostic styles and were separated into 3 main diagnostic style groups. The diagnosis of benign hyperplasia and EC resulted in better overall levels of agreement than EIN. Agreement between the reviewing pathologists and the reference standard (author diagnoses) was not statistically associated pathologist-specific characteristics and working practices. |
| | Atypical hyperplasia/EIN | 0.47 | n.r. | | | | |
| | Endometrial cancer | 0.64 | n.r. | | | | |
| | All diagnostic groups | 0.58 | 79% | | | | |

*(Continued)*

**Table 2.** (Continued)

| Author and year | Interobserver agreement between pathologists | | | Summary of findings | | Comments |
|---|---|---|---|---|---|---|
| | Comparison | Cohens/ Fleiss κ | % Agreement | | | |
| Marotti et al., 2011 [30] | Benign endometrium | n.r. | 67% | **General pathologist characteristics** *Current position of pathologist*: • Surgical pathologist with interest in gynaecologic pathology: n = 32 (41%) • Surgical pathologist with other interests: n = 27 (35%) • Residents/fellows: n = 15 (19%) • Other: n = 4 (5%) *Time in practice, years*: • <3: n = 9 (12%) • 3–10: n = 18 (23%) • >10: n = 36 (46%) • Still in training: n = 15 (19%) *Practice setting*: • Academic: n = 52 (67%) • Private: n = 19 (24%) • Other: n = 7 (9%) *Number of endometrial curettings signed-out per month*: • >20: n = 35 (45%) • 10–20: n = 25 (32%) • <10: n = 18 (23%) *Current terminology used in daily practice*: • WHO criteria: n = 56 (72%) • EIN criteria: n = 20 (26%) • Other: n = 2 (2%) *Pathologist location and use of EIN terminology*: • United States pathologists: (32%) • International pathologists: (3%) | **Pathologist characteristics according to diagnostic agreement** *Current position of pathologist*: • Surgical pathologist with interest in gynaecologic pathology: (62% agreement) • Surgical pathologist with other interests: (57% agreement) • Residents/fellows: (40% agreement) • Other: (47% agreement) *Time in practice, years*: • <3: (49% agreement) • 3–10: (60% agreement) • >10: (61% agreement) • Still in training: (47% agreement) *Practice setting*: • Academic: (56% agreement) • Private: (64% agreement) • Other: (48% agreement) *Number of endometrial curettings signed-out per month*: • >20: (59% agreement) • 10–20: (58% agreement) • <10: (45% agreement) *Current terminology used in daily practice*: • WHO criteria: (58% agreement) • EIN criteria: (57% agreement) • Other: (47% agreement) *Other points*: • 100% agreement was obtained by 2 surgical pathologists with specialised interest in gynaecological pathology • Pathologists who undertook the pre-reading/ training module had higher levels of concordance (60% agreement) compared to those who did not (46% agreement) | Overall agreement between the reviewing pathologists with the reference standard (authors' diagnoses) was largely unaffected by current position, time in practice, number of endometrial curettings signed-out per month, and current terminology used. |
| | Endometrial polyp | n.r. | 35% | | | |
| | EIN | 0.29 | 59% | | | |
| | All groups combined | n.r. | 55% | | | |

*(Continued)*

**Table 2.** (Continued)

| Author and year | Interobserver agreement between pathologists | | | Summary of findings | Comments |
|---|---|---|---|---|---|
| | Comparison | Cohens/ Fleiss κ | % Agreement | | |
| Hecht et al., 2005 [31] | EIN vs non-EIN | 0.54–0.62 | 75% | *Speciality*:<br>• Gynaecological pathologists: n = 3 (100%)<br>*Practice setting*:<br>• Hospital-based working environment: n = 3 (100%)<br>*Main points*:<br>All cases with a *D*-score >1 were accurately classified as benign using subjective criteria. Cases with a *D*-score <1 resulted in more variable interpretation; 27 cases were diagnosed as EIN and 15 as benign (100% sensitivity; 78% specificity)<br>• All future cancer cases occurred in biopsies classed as 'high-risk' both subjectively and morphometrically<br>• Non-EIN diagnosis had a NPV of 100% | The use of the EIN system reduces the likelihood of false positive diagnoses. The subjective application of EIN can work in conjunction with objective morphometry to classify patients into 'high' and 'low' risk groups. |

n.r. = not reported; EIN = endometrioid intraepithelial neoplasia; EH = endometrial hyperplasia; AH = atypical hyperplasia; NPV = negative predictive value;

VPS = volume percentage stroma.

## Pathologist working practices, novel diagnostic methods and interobserver agreement

The main working practices highlighted in the studies are shown in **Table 2**. A common working practice for pathologists is to discuss cases at multidisciplinary meetings and to undertake central pathology reviews. The study by Spoor and Cross in 2019 consisted of a central pathology review, involving seven pathologists reviewing 630 biopsy specimens, which was then compared to the original pathology report [27]. For the diagnosis of atypical hyperplasia, the level of concordance between the central review and the original pathology report was 68%. They then compared the central biopsy review diagnosis with the final hysterectomy diagnosis, and found that even after central review, 23 cases (4.7%) were not correctly classified. Interobserver variation was common when making the distinction between atypical hyperplasia and EC, and in 8 cases (6.1%) EC was diagnosed at biopsy yet not present on hysterectomy, and 47 cases (11.6%) were upgraded from atypical hyperplasia/grade 1 EC to high-grade EC [27].

D'Angelo et al., investigated the agreement between three pathologists when diagnosing EH by using a novel method of grading the "degree" of nuclear atypia, a practice which is not currently included in clinical pathology guidelines [26]. The assessment of the degree of nuclear atypia using a binary classification of "low-grade" and "high-grade" in biopsy specimens resulted in a high level of concordance (87.3–91.1%, κ = 0.72–0.81). When assessing the degree of atypia in hysterectomy specimens, there was reduced concordance amongst the reviewing pathologists (72.2–79.8%, κ = 0.60–0.71). Furthermore, the authors found that the degree of nuclear atypia in biopsy specimens was predictive of the findings at hysterectomy ($p = 1.6 \times 10^{-15}$). However, 16 patients diagnosed with "high-grade" atypical hyperplasia at biopsy were actually found to have grade 1 EC at hysterectomy [26].

Three studies assessed methods of objective diagnosis that could help overcome issues associated with subjective diagnosis [24,25,31]. Sanderson et al., undertook semi-automated computerised image analysis to aid in the quantification of volume percentage stroma (VPS), using 21 consensus EIN (n = 10) and hyperplasia without atypia (n = 11) cases, with the "most abnormal" regions of interest used for the analysis. A VPS of <55% was detected in only 3

cases of EIN, and a VPS >55% was detected in all 11 cases of non-atypical EH [25]. They also determined that immunohistochemical biomarkers such as ARID1A loss ($p$ = 0.011) and altered HAND2 expression ($p$ = <0.001) could aid pathologists in the detection of EIN, and a biomarker panel consisting of HAND2, PTEN and PAX2 could aid in the diagnosis of benign EH. The '$D$-score' is the morphometrical analysis of endometrial gland architecture and cytology, and a $D$-score <1 means EIN should be diagnosed, whereas a '$D$-score' >1 means the endometrial lesion is benign or non-atypical EH [8]. In the study by Hecht et al., they found that all cases with a $D$-score >1 were correctly diagnosed as benign (100% specificity), however the sensitivity was only 78% as 15 cases with a D-score <1 were diagnosed as benign [8,31]. In the study by Zhao et al., a global (cytological changes in lesion background)-to-local (gland-to-stroma ratio, lesion dimensions) multi-scale convolutional neural network (G2LNet) was developed [24]. Two specialist pathologists (over 20 years endometrial pathology experience) labelled all haematoxylin and eosin (H&E) slides and divided them into "normal endometrium", "hyperplasia without atypia" and "EIN". In an external validation dataset of 1631 H&E images, the G2LNet deep learning model achieved an overall accuracy of 95.3% (95% CI: 94.3–96.4%), which was higher than the junior pathologist with 85% (95% CI: 83.3–86.8%), but not as good as the senior pathologist who had an accuracy of 98.7% (95% CI: 98.1–99.2%). Kappa values between the three reviewing pathologists ranged from 0.775 to 0.9732, and the senior pathologist had the highest Kappa agreement with G2LNet (κ = 0.93) [24].

## Additional outcomes of interest

Variation between diagnostic groups and intra-observer variation were additional outcomes of interest. Six studies provided the Cohen's κ or percentage agreement for different diagnostic groups, such as benign endometrium, atypical hyperplasia/EIN, and endometrial cancer, **Table 2** [24–26,28–30]. The diagnosis of atypical hyperplasia/EIN resulted in the worst overall levels of agreement, ranging from 7.1% - 98.1% agreement across the six studies. In the study by Zhao et al., the percentage agreement for the diagnosis of a normal proliferative endometrium was highest at 99.4–100%, compared to 82.8–97.8% for hyperplasia without atypia and 74.9–98.1% for EIN between the three reviewing pathologists with differing levels of experience [24]. In the same study, the G2LNet deep learning model accurately diagnosed 98.3% of normal endometrium, 85.8% of hyperplasia without atypia and 99.8% of EIN cases–the latter outperforming the reviewing pathologists [24]. Ordi et al., observed higher agreement levels when diagnosing benign cycling endometrium and EC, compared to benign hyperplasia and EIN [28], with Usubutun et al., concluding that the diagnosis of EIN resulted in poorer reproducibility compared to benign hyperplasia and EC [29].

Two studies additionally assessed intraobserver outcomes, see **S1 Table.** Hecht et al., investigated the presence or absence of EIN on two separate occasions (timeframe not specified), and reported that intraobserver variation was very good, with an overall reproducibility of 92.8% [31]. Similarly, D'Angelo et al., reported high level of intraobserver agreement when grading the degree of nuclear atypia in both biopsy (96.2%– 97.5%) and hysterectomy specimens (89.9%– 98.7%) [26]. In their study, the histological slides were re-reviewed by the pathologists after a two-month interval.

## Discussion

Interobserver variability amongst pathologists in the diagnosis of EH is well-documented, particularly in relation to the main histopathological factors that contribute to diagnostic discordance. However, the influence of pathologist-specific characteristics and/or working practices on interobserver variation in the diagnosis of EH is less well known, and to the best of our

knowledge, this systematic review is the first to examine these factors. Eight studies were included in this review, and the main findings demonstrated that pathologists have different diagnostic styles, with some more likely to under-diagnose and some more likely to over-diagnose endometrial lesions. Some novel working practices were identified that are not currently recommended in clinical guidelines, such as grading the "degree" of nuclear atypia, and the incorporation of objective methods of diagnosis such as semi-automated quantitative image analysis/deep learning models. These methods resulted in reproducible EH diagnoses, although more research is warranted to determine their applicability in clinical settings.

Overall, this systematic review found little evidence that pathologist-specific factors influenced interobserver variation in the diagnosis of EH including current position, time in practice, number of endometrial biopsies signed-out per month, or the classification criteria used in daily practice. However, only two studies investigated pathologist factors in significant detail [29,30]. An interesting observation was that pathologists could be grouped according to how they make a diagnosis [29]. Different diagnostic styles was also evident across a number of the included studies. For example, one pathologist in the study by Sanderson et al. diagnosed 20 more cases of EIN than the second reviewing pathologist, which may suggest that the pathologists had different diagnostic styles, with one more likely to over-diagnose [25]. In addition, this reduced concordance in the diagnosis of the atypical lesion was also observed in the other studies, as the diagnosis of benign endometrium, hyperplasia without atypia or EC resulted in higher levels of agreement than the diagnosis of atypical hyperplasia/EIN [24,28–30]. It might have been expected that the level of pathologist experience, i.e., time in practice or general pathologists compared to specialist gynaecological pathologists, might be a source of potential interobserver variation and there was some evidence of this in this systematic review [24]. However, other studies demonstrated that specialist gynaecological pathologists and general pathologists had similar levels of agreement with the reference standard [29,30], although both these studies included pre-training and reading material that may have resulted in increased concordance. Nonetheless, in some studies that included only specialist gynaecological pathologists [25,28], interobserver variation was still prominent, suggesting that even pathologists with extensive years of specialist experience are still subject to differences in diagnostic opinion. Accurately quantifying the extent of interobserver variation is a difficult process, and it is possible that the high levels of interobserver variability in the included studies could in part be explained by the study protocols. These protocols are not reflective of typical, everyday working practices where prior and additional endometrial biopsies can be reviewed, clinical (including patient age) and radiological information is accessible, and colleagues can be consulted [30,32].

In 2020, Cancer Research UK reported that without intervention, the number of histopathologists in the UK is expected to reduce by 2% by 2029 [33]. This shortage of pathologists is occurring on a global scale, with the number of new pathologists declining at a steady rate [34–36]. It is therefore essential that resources and working practices are optimised whilst also ensuring diagnostic accuracy. This review highlights that novel methods such as assessing the degree of nuclear atypia [26], and the incorporation of objective approaches, deep learning models and/or biomarker panels can potentially assist pathologists in their diagnosis [24,25,31]. Classification of atypical hyperplasia on endometrial biopsy into low-grade and high-grade atypical hyperplasia, using a binary classification of nuclear atypia, was shown to be reproducible amongst gynaecological pathologists from three different institutions [26]. Specific cytological criteria were outlined in the study for both "high-grade" and "low-grade" nuclear atypia, and architectural complexity was also considered. None of the cases of low-grade atypical hyperplasia were associated with carcinoma on subsequent hysterectomy, while in 61% of high-grade atypical hyperplasia's on biopsy, there was a grade 1 endometrioid

carcinoma in the hysterectomy specimen. However, this study included only 79 patients and three pathologists. Therefore, future studies are required in larger patient populations to assess wider reproducibility and to determine if this is a feasible method for incorporation into diagnostic criteria. Furthermore, assessing the degree of nuclear atypia in biopsy specimens was more reproducible amongst the reviewing pathologists than in the hysterectomy specimens. This may be attributed to the fact that in the biopsy specimens, the pathologists only had two diagnostic categories ("low" or "high-grade" atypia) and in the hysterectomy specimens, there were more diagnostic categories (see **Table 2**), which may have resulted in increased discordance. Some of the included studies assessed both biopsy and hysterectomy specimens, which are two very different specimen types. This may mean it can be difficult to make adequate comparisons in terms of interobserver variation. In addition, many studies included patients who had undergone both endometrial biopsy and hysterectomy, and so further studies specific to endometrial biopsy specimens are required to identify ways of improving diagnostic accuracy of EH. This is especially relevant for EH patients diagnosed with hyperplasia without atypia, or for patients who may not undergo hysterectomy immediately due fertility preservation wishes or for whom surgery in contraindicated due to significant comorbidity. A study by Bryant et al in 2019 which did not meet our systematic review criteria identified that selective review of hysterectomy specimens (i.e., review of every other pathology block) could have potential as a reproducible method that could reduce costs and assist in a declining pathology workforce [37]. However, this method resulted in some cases of atypical hyperplasia being diagnosed as benign, which could have negative consequences for patient management and lead to the potential under- and over-treatment of patients. Therefore, it is currently unlikely that selective review could be routinely implemented, although further large-scale studies are warranted.

A number of studies aimed to investigate more objective methods of EH diagnosis. Computer-aided imaging of gland-to-stroma ratio in EH specimens was found to assist pathologists in improving diagnostic accuracy and was useful in the exclusion of an atypical hyperplasia/EIN diagnosis, as all cases of non-atypical EH were successfully detected using a VPS >55% [25]. However, the study was based on a very small set of only 21 EH specimens, and so larger studies are required to validate this method as a potential diagnostic tool. The use of objective computerised morphometry by Hecht et al., was shown to aid classification of patients into 'high' and 'low' risk groups that could predict progression to EC, although specificity was only 78% [31]. Furthermore, the *D*-score calculation may not be widely applicable in routine practice as it requires the use of costly equipment and highly experienced pathologists, whilst also being highly time-consuming [16]. A deep learning model termed 'G2LNet' was shown to perform better than three reviewing pathologists in identifying atypical hyperplasia/EIN; however, the pathologists performed better than G2LNet in identifying cases of normal endometrium and hyperplasia without atypia [24]. Overall, the authors found that G2LNet was superior or comparable to a junior (two-years' experience) and mid-level pathologist (six-years' experience), highlighting the potential of such methods in reducing the diagnostic burden on pathologists. The study however, did not assess the ability of G2LNet to distinguish between atypical EH and EC, and the model struggled in the accurate interpretation of diagnostic features in images with fragmentation [24]. Furthermore, consensus cases were used in place of diagnostically difficult cases, which may have increased the performance of the model. Nonetheless, future research could investigate the potential of this innovative tool in assisting pathologists with diagnostically difficult cases, as well as its utilisation as a screening tool for triaging patients with a suspected diagnosis of EH [24].

In the most recently published 2020 WHO criteria for the diagnosis of EH, the use of biomarkers was as an adjunct to diagnosis of atypical hyperplasia is listed as "desirable" criteria

[6]. The suggested biomarkers are PTEN, PAX2 and mismatch repair proteins. Sanderson et al., found that altered HAND2 expression and loss of ARID1A were significantly associated with a diagnosis of atypical hyperplasia/EIN, and a biomarker panel of HAND2, PTEN and PAX2 was able to identify those likely to have hyperplasia without atypia [25]. However, further research is warranted to determine the applicability of this panel in the diagnostic setting for EH. Interestingly, in a recently conducted international survey of pathologist working practices in the diagnosis of post-hormonal therapy EH specimens, 76% of 95 responding pathologists reported that they never undertook immunohistochemical staining [32]. This is despite significant advancements in the field regarding the immunohistochemical profile of EH, including proposed biomarker panels for the diagnosis of atypical hyperplasia/EIN [38]. In addition, a number of novel diagnostic innovations utilising biomarkers detected in blood, urine, and cervico-vaginal fluid are currently under investigation for the early detection of EC, but there has been limited investigation in EH [39]. More research is needed to help determine whether biomarker panels and image analysis methods are feasible in everyday practice for the diagnosis of EH.

Our systematic review has a number of strengths. We investigated recent trends in interobserver variation by restricting eligible studies to those using updated classification criteria for EH diagnosis. We used a broad search strategy and conducted independent screening of articles, as well as undertaking a thorough quality assessment of included studies. However, the methodologies used to quantify interobserver variation varied across the studies, which made it difficult quantify the extent of reproducibility. Furthermore, most studies very limited detail on pathologist-specific characteristics, with small sample sizes and few reviewing pathologists, so true reproducibility was hard to determine from such limited analyses and thus, may not be reflective of everyday working practices. Although only recently published, none of the studies used the updated 2020 WHO criteria, including the use of biomarkers, for the diagnosis of EH, so it remains unclear whether this update could help overcome current reproducibility issues (through the increased use of biomarkers). Despite these limitations, the findings from this systematic review provides an insight into current pathologist working practices in the diagnosis of EH and could help inform future clinical guidelines to improve diagnostic accuracy.

## Conclusions

In summary, this systematic review highlights the most recent trends in interobserver variation and pathologist working practices in the diagnosis of EH and identified some possible approaches to improve diagnostic concordance. Grading the degree of nuclear atypia and the incorporation of objective methods of diagnosis such as computer-aided imaging and deep learning models resulted in reproducible diagnoses, although most of the studies were small with few reviewing pathologists. Furthermore, the applicability of these methods in routine everyday practice is unknown. Future research efforts should investigate if incorporating such working practices, including biomarker panels, is feasible and reproducible on a widespread scale, with the ultimate aim of informing pragmatic interventions that could minimise variation in diagnostic reporting of EH, and therefore, optimising patient care.

## Supporting information

**S1 Checklist. PRISMA-P checklist.**
(DOCX)

**S1 Table. Intraobserver variation outcomes in the diagnosis of endometrial hyperplasia in two of the included studies.**
(DOCX)

**S1 Appendix. Search strategy for MEDLINE and EMBASE.**
(DOCX)

**S2 Appendix. Search strategy for web of science.**
(DOCX)

## Author Contributions

**Conceptualization:** Helen G. Coleman.

**Data curation:** Chloe A. McCoy, Helen G. Coleman, Charlene M. McShane, Úna C. McMenamin.

**Formal analysis:** Chloe A. McCoy, Helen G. Coleman, Charlene M. McShane.

**Funding acquisition:** Úna C. McMenamin.

**Investigation:** Chloe A. McCoy, Helen G. Coleman, Charlene M. McShane, Úna C. McMenamin.

**Validation:** W. Glenn McCluggage, James Wylie.

**Visualization:** Chloe A. McCoy, Helen G. Coleman, Úna C. McMenamin.

**Writing – original draft:** Chloe A. McCoy.

**Writing – review & editing:** Chloe A. McCoy, Helen G. Coleman, Charlene M. McShane, W. Glenn McCluggage, James Wylie, Declan Quinn, Úna C. McMenamin.

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
