## [Decision Letter · Decision Letter 0]

1 Feb 2024

PONE-D-23-40071Factors associated with interobserver variation amongst pathologists in the diagnosis of endometrial hyperplasia: a systematic reviewPLOS ONE

Dear Dr. McCoy,

Thank you for submitting your manuscript to PLOS ONE. After careful consideration, we feel that it has merit but does not fully meet PLOS ONE’s publication criteria as it currently stands. Therefore, we invite you to submit a revised version of the manuscript that addresses the points raised during the review process.

**The paper was reviewed by two independent reviewers highly expert in the field and their comments and suggestions are appended below. You need to address each inquires identified during the review process.**

We look forward to receiving your revised manuscript.

Kind regards,

Asmerom Tesfamariam Sengal, MD, PhD

Academic Editor

PLOS ONE

Journal Requirements:

Reviewers' comments:

Reviewer's Responses to Questions

**Comments to the Author**

1. Is the manuscript technically sound, and do the data support the conclusions?

Reviewer #1: Yes

Reviewer #2: Yes

2. Has the statistical analysis been performed appropriately and rigorously? 

Reviewer #1: N/A

Reviewer #2: Yes

3. Have the authors made all data underlying the findings in their manuscript fully available?

Reviewer #1: Yes

Reviewer #2: Yes

4. Is the manuscript presented in an intelligible fashion and written in standard English?

Reviewer #1: Yes

Reviewer #2: Yes

5. Review Comments to the Author

Reviewer #1: Thank-you for the opportunity to review this interesting and well written paper. This is a review of published papers that specifically includes those with WHO (2014) or EIN based diagnosis of endometrial hyperplasia and also includes information on interobserver variability in diagnosis.

I do have a few comments and questions about the paper as it is currently presented.

page 2 - objective and page 4 line 92-94: The objectives do not read as relating solely to pathologist-specific factors (ie not including specimen related factors) - suggest tightening wording of objective.

page 2 line 42-44 Conclusion - first / second sentence of conclusion reads somewhat clunky to me. Direct line between objective of study and conclusion could be clearer.

page 13 table 2 - Zhao et al - comments section missing a word/s

page 14 table 2 - spoor et al - I personally draw a distinction between "not malignant" and "no residual malignancy" and from my reading of the paper it is that latter the authors are documenting.

page19 line 237-239 sentence not clear to reading.

page 22 lines from 318 to 326 : I think review of this paragraph(s) would improve the readability. The text seems to contradict itself and the overall message is not clear.

page 22 line 334 - 337 Agree, and including further levels

Fig 1 - not clear enough in my version (online or downloaded).

Overall I think this is an interesting and informative paper.

Reviewer #2: This is a well developed and thoroughly researched contribution. The author's emphasis on clinical utility is appreciated. The state of the field, currently in flux, is well assessed and the author is very forward looking. It's also well written - complex ideas are well and clearly presented.

A few minor comments and suggestions for minor revision:

- the authors emphasize the importance of accurate and early detection of EH for appropriate management in their opening paragraph, and also discuss the issues of working with biopsy/curettage specimens (inadequacy, fragmentation, focality). However many of the studies include both biopsy and hysterectomy specimens - i.e. some of these patients are already "managed", and these are two very different specimen types when it comes to assessment, each with their own challenges and issues. I think it's reasonable should be commented on at some point in the discussion.

-Spoor and Cross's paper is a great paper for validating the role of central review and MDTs. The figures used in table 2 regarding discrepant results should include percentages - the 47 upgrades is somewhat shocking, without the context of the relatively large number of cases and other details in the paper.

6. PLOS authors have the option to publish the peer review history of their article (what does this mean?). If published, this will include your full peer review and any attached files.

Reviewer #1: No

Reviewer #2: No

---

## [Author Response · Author response to Decision Letter 0]

11 Mar 2024

Reviewer comments

Reviewer #1: Thank-you for the opportunity to review this interesting and well written paper. This is a review of published papers that specifically includes those with WHO (2014) or EIN based diagnosis of endometrial hyperplasia and also includes information on interobserver variability in diagnosis. I do have a few comments and questions about the paper as it is currently presented.

* We would like to thank the reviewer for their helpful and constructive feedback, and have responded to each of their queries below:

page 2 - objective and page 4 line 92-94: The objectives do not read as relating solely to pathologist-specific factors (ie not including specimen related factors) - suggest tightening wording of objective.

* We agree that it should be emphasised that pathologist-specific factors is what we were investigating, and this has been amended accordingly 

(Page 2, lines 26-27; Page 4, lines 93-94: “This systematic review aimed to identify pathologist-specific factors associated with interobserver variation in the diagnosis and reporting of EH”).

Page 2 line 42-44 Conclusion - first / second sentence of conclusion reads somewhat clunky to me. Direct line between objective of study and conclusion could be clearer.

* We have specified that our review highlighted the impact of pathologist-specific factors and working practices in the accurate diagnosis of endometrial hyperplasia 

(Page 2, lines 42-48: “This review highlighted the impact of pathologist-specific factors and working practices in the accurate diagnosis of EH, although few studies have been conducted. Further research is warranted in the development of more objective criteria that could improve reproducibility in EH diagnostic reporting, as well as determining the applicability of novel methods such as grading the degree of nuclear atypia in clinical settings”.

Page 13 table 2 - Zhao et al - comments section missing a word/s

* We have added in the word “used” to make the comment section for Zhao et al clearer. 

Page 14 table 2 - spoor et al - I personally draw a distinction between "not malignant" and "no residual malignancy" and from my reading of the paper it is that latter the authors are documenting.

* We have replaced ‘not malignant’ with ‘there was no cancer present’ in the table, based on the wording in the discussion of the Spoor and Cross paper.

Page 19 line 237-239 sentence not clear to reading.

* The wording of the results has been amended to reflect the difference more accurately between the central biopsy review diagnosis and the final hysterectomy diagnosis

(see Page 19, lines 235-240 – “For the diagnosis of atypical hyperplasia, the level of concordance between the central review and the original pathology report was 68%. They then compared the central biopsy review diagnosis with the final hysterectomy diagnosis, and found that even after central review, 23 cases (4.7%) were not correctly classified.”

Page 22 lines from 318 to 326 : I think review of this paragraph(s) would improve the readability. The text seems to contradict itself and the overall message is not clear.

* This has been reworded to emphasise that atypical hyperplasia resulted in the most diagnostic disagreements, and one pathologist in the study by Sanderson et al diagnosed 20 more cases of atypical hyperplasia/EIN than the second reviewing pathologist, which may suggest that they had different diagnostic styles

(See Page 22, lines 321-326: “For example, one pathologist in the study by Sanderson et al. diagnosed 20 more cases of EIN than the second reviewing pathologist, which may suggest that the pathologists had different diagnostic styles, with one more likely to over-diagnose25. In addition, this reduced concordance in the diagnosis of the atypical lesion was also observed in the other studies, as the diagnosis of benign endometrium, hyperplasia without atypia or EC resulted in higher levels of agreement than the diagnosis of atypical hyperplasia/EIN 24, 28-30”.

page 22 line 334 - 337 Agree, and including further levels

* We have added in that additional endometrial biopsies that may be required could also be reviewed, which is not reflected in the study protocols

(See Page 23, lines 339-341: “These protocols are not reflective of typical, everyday working practices where prior and additional endometrial biopsies can be reviewed, clinical (including patient age) and radiological information is accessible, and colleagues can be consulted”.

Fig 1 - not clear enough in my version (online or downloaded).

* Thank you for highlighting this, we have increased the DPI/resolution of both Figure 1 and Figure 2, and hopefully they are now clearer. 

Reviewer #2: This is a well developed and thoroughly researched contribution. The author's emphasis on clinical utility is appreciated. The state of the field, currently in flux, is well assessed and the author is very forward looking. It's also well written - complex ideas are well and clearly presented.

* We thank the reviewer for their useful feedback and careful analysis of our work. Please find responses to your queries below:

A few minor comments and suggestions for minor revision:

- the authors emphasize the importance of accurate and early detection of EH for appropriate management in their opening paragraph, and also discuss the issues of working with biopsy/curettage specimens (inadequacy, fragmentation, focality). However many of the studies include both biopsy and hysterectomy specimens - i.e. some of these patients are already "managed", and these are two very different specimen types when it comes to assessment, each with their own challenges and issues. I think it's reasonable should be commented on at some point in the discussion.

* This is a very good point and we have commented on this in the discussion:

(Page 23, lines 364-372: “Some of the included studies assessed both biopsy and hysterectomy specimens, which are two very different specimen types. This may mean it can be difficult to make adequate comparisons in terms of interobserver variation. In addition, many studies included patients who had undergone both endometrial biopsy and hysterectomy, and so further studies specific to endometrial biopsy specimens are required to identify ways of improving diagnostic accuracy of EH. This is especially relevant for EH patients diagnosed with hyperplasia without atypia, or for patients who may not undergo hysterectomy immediately due fertility preservation wishes or for whom surgery in contraindicated due to significant comorbidity.”

-Spoor and Cross's paper is a great paper for validating the role of central review and MDTs. The figures used in table 2 regarding discrepant results should include percentages - the 47 upgrades is somewhat shocking, without the context of the relatively large number of cases and other details in the paper.

* We agree, and we have added in the percentages for clarity – both in Table 2 (Page 14) and in the text where these results are discussed (Page 19, lines 242-243) .

---

## [Decision Letter · Decision Letter 1]

1 Apr 2024

Factors associated with interobserver variation amongst pathologists in the diagnosis of endometrial hyperplasia: a systematic review

PONE-D-23-40071R1

Dear Dr. McCoy,

We’re pleased to inform you that your manuscript has been judged scientifically suitable for publication and will be formally accepted for publication once it meets all outstanding technical requirements.

Kind regards,

Asmerom Tesfamariam Sengal, MD, PhD

Academic Editor

PLOS ONE

Additional Editor Comments (optional):

Reviewers' comments:

Reviewer's Responses to Questions

**Comments to the Author**

1. If the authors have adequately addressed your comments raised in a previous round of review and you feel that this manuscript is now acceptable for publication, you may indicate that here to bypass the “Comments to the Author” section, enter your conflict of interest statement in the “Confidential to Editor” section, and submit your "Accept" recommendation.

Reviewer #1: All comments have been addressed

2. Is the manuscript technically sound, and do the data support the conclusions?

Reviewer #1: Yes

3. Has the statistical analysis been performed appropriately and rigorously? 

Reviewer #1: N/A

4. Have the authors made all data underlying the findings in their manuscript fully available?

Reviewer #1: Yes

5. Is the manuscript presented in an intelligible fashion and written in standard English?

Reviewer #1: Yes

6. Review Comments to the Author

Reviewer #1: Dear Authors,

Thank-you for the opportunity to review this article following the changes made in response to the previous reviews. Reading it afresh it was clearer, I feel the changes you made in response to the comments have improved the communication of the message.

I did have a few small things I noticed this time around for your consideration.

Page 3 Line 58- 59: I found this slightly confusing to read due to the use of / which is also use to denote two terms with the same meaning (which is the way you have used it in the abstract page 2 line 40-41.

Page 24 lines 371-374 : Perhaps clarify this reference of selective review to refer to hysterectomies, not biopsies.

Thanks again, a well written and interesting paper.

7. PLOS authors have the option to publish the peer review history of their article (what does this mean?). If published, this will include your full peer review and any attached files.

Reviewer #1: No
